# Molecular Classifiers in Skin Cancers: Challenges and Promises

**DOI:** 10.3390/cancers15184463

**Published:** 2023-09-07

**Authors:** Ali Azimi, Pablo Fernandez-Peñas

**Affiliations:** 1Westmead Clinical School, Faculty of Medicine and Health, The University of Sydney, Westmead, NSW 2145, Australia; 2Department of Dermatology, Westmead Hospital, Westmead, NSW 2145, Australia; 3Centre for Cancer Research, The Westmead Institute for Medical Research, The University of Sydney, Westmead, NSW 2145, Australia

**Keywords:** skin cancer, molecular classifier, melanoma, basal cell carcinoma, cutaneous squamous-cell carcinoma

## Abstract

**Simple Summary:**

Skin cancers are common and sometimes difficult to diagnose malignancies that occur worldwide. Most skin cancers are diagnosed by visual assessment of patient samples obtained through biopsy. However, due to the lack of well-defined malignancy features, the diagnosis and classification of skin cancer lesions remain difficult in some cases. To overcome this issue, researchers have attempted to use molecular information such as genes and proteins and imaging data to improve skin cancer diagnosis and classification. Therefore, this paper reviews the recent advancements in large-scale molecular profiling approaches and appraises their limitations and potential for reliable and reproducible classification and stratification of skin cancers.

**Abstract:**

Skin cancers are common and heterogenous malignancies affecting up to two in three Australians before age 70. Despite recent developments in diagnosis and therapeutic strategies, the mortality rate and costs associated with managing patients with skin cancers remain high. The lack of well-defined clinical and histopathological features makes their diagnosis and classification difficult in some cases and the prognostication difficult in most skin cancers. Recent advancements in large-scale “omics” studies, including genomics, transcriptomics, proteomics, metabolomics and imaging-omics, have provided invaluable information about the molecular and visual landscape of skin cancers. On many occasions, it has refined tumor classification and has improved prognostication and therapeutic stratification, leading to improved patient outcomes. Therefore, this paper reviews the recent advancements in omics approaches and appraises their limitations and potential for better classification and stratification of skin cancers.

## 1. Introduction

Currently, clinical data combined with imaging technologies such as digital dermoscopy, confocal microscopy, multiphoton microscopy and optical coherence tomography are used to diagnose and observe changes in skin cancers over time [1,2]. Suspected lesions are subsequently excised for histopathology analysis to confirm or rule out invasive tumors. However, due to the lack of clear-cut discriminatory features, clinical and imaging false-positive and false-negative diagnoses of skin cancers and their premalignant lesions are common with undesirable consequences for the patient and the healthcare system [3,4,5,6,7]. For example, the number of benign excisions for every melanoma diagnosis ranges from 3:1 in highly specialist dermatology settings to 8:1 in general dermatology and 30:1 in primary care [5,6]. Considering non-melanoma skin cancers (NMSCs), cutaneous squamous-cell carcinomas (cSCCs) are reported to be clinically misdiagnosed as basal-cell carcinomas (BCCs) and vice versa, with a study reporting 13 out of 154 cases inversely diagnosed [8].

Moreover, it is well recognized that due to considerable morphologic heterogeneity, the lack of clear-cut features of malignancy and interobserver disagreements, the diagnosis of skin cancers can sometimes be complicated by the pathologic assessment of a biopsy. A 7-year single-center analysis showed that from a total of 936 histologically confirmed melanomas, 16% were misdiagnosed, of which 46.7% were pigmented nevus, 21.3% were BCCs, 10.7% were unspecified tumors, 6% were dysplastic nevus and 3.3% were cSCC [9]. Similarly, histopathology assessment fails to correctly identify around 30% of primary cSCC tumors with no defining histological features that may carry the risk of progression to regional lymph nodes or distant body organs, while 75% of the lesions identified with a risk of metastasis do not develop metastasis [10]. A BCC biopsy sample may also lack aggressive components, and its subtyping is subject to interobserver variation [11]. It is reported that a maximum of 60.9% agreement can be achieved between punch biopsy and subsequent surgical excision analysis of primary BCCs, indicating limitations of a histopathological assessment of punch biopsies [12].

Considering melanocytic skin lesions, a 2017 study also showed that benign nevi and advanced melanomas had 92% and 72% diagnostic and classification accuracy using histopathological criteria, respectively. On the other hand, lesions classified in between, such as dysplastic nevi, melanoma in situ, and early invasive melanoma, were properly diagnosed in 25%, 40% and 43% of the cases [13].

Since therapeutic interventions in skin cancers are primarily based on a broad clinical and histopathological diagnosis, their misclassification and misdiagnosis will lead to ineffective, inappropriate or delayed treatment. Despite recent improvements and access to novel therapies, such as checkpoint inhibitor immunotherapy and chemotherapy, a significant number of patients with advanced skin carcinomas die every year, and there is no personalized platform to stratify and match patients to the most effective therapeutic agents based on the molecular composition and profile of their tumors [14,15].

## 2. Classification of Skin Cancers

Skin cancer is a complex and multifactorial disease that involves changes in multiple levels of biological information, including genes, transcripts, proteins, and metabolites that produce phenotypic and physical changes on dermoscopy, confocal microscopy, and histology, as well as clinical manifestations. Changes in the characteristics of each layer of information, either alone or in combination with other layers provide an opportunity for a better diagnosis and classification of skin cancers (Figure 1).

At the molecular level, genetic and proteomic changes in skin cancers are numerous and varied. Changes in the expression, interaction, and other alterations in various genes and proteins have been associated with skin cancers, including in DNA repair, cell cycle regulation and tumor suppression [16,17,18,19]. On the other hand, metabolites are involved in various biochemical processes within skin cancer cells, including energy production, signaling and regulation of cellular processes. Changes in the levels, composition, interaction and structure of metabolites can have profound effects on cellular function and can contribute to the development of various skin cancers including melanoma [20] and cSCCs [21]. Altogether, these molecular changes can lead to alterations in the expression of various phenotypes, which may contribute to the development and progression of skin cancers.

Histological examination of skin biopsies together with clinical assessment such as location and size of the lesion, as well as patient demographics and risk factors, remains the gold standard for diagnosing and classification of skin cancers. Histology can provide information on the type and stage of skin cancers, as well as the presence of important features such as mitotic rate, tumor thickness, level of differentiation, depth of invasion, etc. Classically, melanoma, cSCC and BCCs are understood to evolve from normal skin cells, progressing through premalignant stages (excluding BCC), before entering the in situ phase, where the lesions are confined to the epidermis layer of the skin. The accumulation of additional mutations in carcinogenic genes, sometimes influenced by hereditary factors, tissue microenvironment and environmental triggers, particularly UV exposure, can propel the tumor’s advancement. This leads to the invasion of the dermis layer and, subsequently, the potential spread to nearby lymph nodes and distant organs. Each of these stages exhibits distinct clinical and histological characteristics, along with varying levels of survival rates and treatment responses. A condensed overview of the traditional stages and progression of cutaneous skin cancers, based on TNM staging and the involvement of key genes in disease advancement, is depicted in Figure 2 (melanoma) and Figure 3 (cSCC and BCC).

Imaging features, such as dermoscopy and reflectance confocal microscopy have also provided important diagnostic and classification information. Dermoscopy is a non-invasive imaging technique that allows for the visualization of skin lesions at high magnification. Reflectance confocal microscopy provides cellular-level imaging of skin lesions and can be used to distinguish between benign and malignant lesions. Recently, the application of artificial intelligence (AI) on dermoscopy and confocal microscopy images of skin cancers, particularly melanoma, has been shown to perform similarly to and on some occasions better than experienced dermatologists for tumor diagnosis and classification [22,23,24]. However, the use of AI in skin cancer diagnosis, classification and risk assessment is yet to be implemented in the clinical setting.

However, skin cancers’ development and progression, resistance to therapy and changes in the tumor during treatment embody alterations in the genomic, transcriptomic and proteomic landscape of cancer cells and their microenvironment, which cannot be grasped by the assessment of tumors using traditional measures such as tumor stages, grading and phenotypic presentations alone [25,26]. Therefore, a reliable molecular approach to tumor classification or “molecular classifier” is needed to overcome these challenges.

Summarizing the clinical, imaging and histological classification of skin cancers is beyond the scope of the present study. Thus, the purpose of this review is to examine molecular classifiers in three prevalent types of skin cancers: melanoma, cSCC and BCC. Specifically, it will explore the utilization of gene and protein classifiers that have the potential to assist in diagnosing or predicting clinical behaviors such as prognosis, survival, metastasis and response to therapy in these skin cancers.

**Figure 3 cancers-15-04463-f003:**
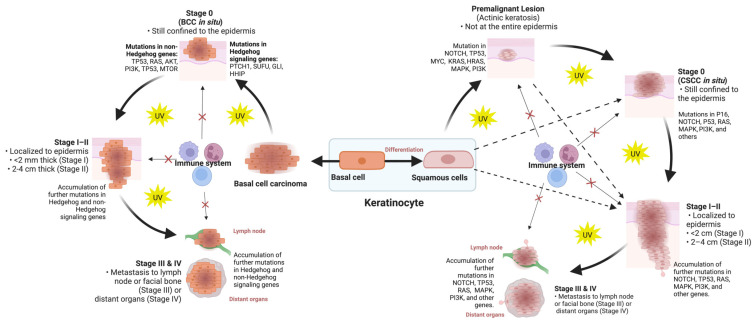
Overview of cSCC and BCC development and progression from keratinocytes. This figure illustrates the sequential stages of melanoma development and progression. Cutaneous squamous-cell carcinoma (cSCC) and basal-cell carcinomas (BCCs) stem from keratinocytes, the principal cells of the skin. Aberrant keratinocyte growth initiates these skin cancers. Within the skin, basal cells can differentiate into squamous cells or remain in the basal layer, dividing as basal cells. Like melanomas, cSCC and BCCs consist of four stages based on the TNM staging system. cSCC’s progression commences with normal keratinocytes, advancing through premalignant stages like actinic keratosis and Bowen’s disease. Specific gene mutations drive this process. Factors like UV light exposure, immune-related concerns including immunosuppression and other environmental influences cause genomic and proteomic changes, enabling tumor invasion and metastasis to lymph nodes and distant organs. Notably, less than 0.6% of actinic keratosis cases may progress to cSCC, with approximately 70% of cSCCs originating from actinic keratosis [27]. For BCCs, a similar TNM staging applies, yet a premalignant lesion is yet to be identified. Mutations in Hedgehog and non-Hedgehog pathway genes, the immune microenvironment, sun exposure and other factors significantly contribute to BCC development and progression, constructing the intricate molecular landscape of these skin malignancies. × indicates inactivation or disruption of the immune system.

## 3. Molecular Classification of Skin Cancers

Recent advancements in large-scale “omic” studies, including genomics, transcriptomics, proteomics, metabolomics and imaging-omics, have provided invaluable information about various human cancers’ molecular and visual landscapes. In combination with machine learning (ML) approaches, omics have significantly improved our understanding of human tumors, including those arising in the skin, offering limitless potential for precision medicine and innovations in clinical management [28,29]. Omic analyses have successfully identified new and unified classes or sub-classes of cancers. The largest of its kind, a genomic and proteomic study of 3527 specimens from 12 cancer types, including head and neck SCC, breast cancer and colon cancer, by the Cancer Genome Atlas Research Network classified the tumors into 11 major subtypes that correlated with the tissue of origin [29]. For example, distinct cancers such as lung, head and neck, and a subset of bladder squamous tumors were grouped into one subtype based on their similar level of TP53 alterations, TP63 amplifications and immune/proliferation pathway gene elevations.

On the other hand, based on their unique molecular composition, bladder cancers were split into three pan-cancer subtypes. Other studies have also explored various molecular classification approaches in tumors such as breast [30], endometrial [31], bladder [32] and many more cancers, leading to the identification of new molecular classes of the diseases with distinct outcomes. These refined molecular classifications of tumors facilitated by omic data provide the potential for improved detection, management and novel therapeutic strategies.

### 3.1. Genomic and Transcriptomic Classification in Skin Cancers

Like other tumors, inherited mutations or a gradual accumulation of somatic gene alterations permit the emergence of neoplastic properties in skin cells, leading to the progression from normal skin to premalignant lesions, which eventually transform into advanced and aggressive skin cancers. Germline and somatic mutation analysis of skin cancers has been studied extensively in the past few years, providing novel information about the genome landscape of these lesions and identifying driver genes and critical molecular players in their initiation and progression [33,34,35,36,37]. A summary of a number of genomic and transcriptomic analyses identifying novel classes or groups of melanomas, cSCC and BCCs with the potential to aid in diagnosis of melanoma is presented in Table 1a–c.

#### 3.1.1. Cutaneous Melanoma

The approach for classifying cutaneous melanoma is primarily based on clinical and histopathological criteria. However, genomic and transcriptomic changes alone or in combination with clinical and histological criteria are increasingly used for staging and classifying lesions. Genomic profiling of advanced melanoma has improved its diagnosis and prognosis. For example, molecular testing for BRAF [56], NRAS mutations [18,19] and immunostainings against P16, PRAME [18], S100, Melan A, Ki67 and MITF proteins [19] are found to offer excellent potential for the diagnosis of melanoma, which otherwise would not have been possible using histopathology assessment alone.

Due to their importance for a better and more comprehensive classification, in 2018 the World Health Organization widely incorporated genomic changes into the classification of melanomas [38]. Based on the study by Elder et al. [38], this updated classification system recognizes nine evolutionary pathways to melanoma development which includes superficial spreading melanoma in Pathway I, lentigo malignant melanoma in Pathway II and desmoplastic melanoma in Pathway III, which all three pathways typically associated with cumulative solar damage (CSD). Pathways IV to IX which are not associated with CSD includes spitz melanoma, acral melanoma, mucosal melanoma, melanoma arising in congenital nevi, melanoma arising in blue nevi and uveal melanoma, respectively. On the other hand, nodular melanoma is classified to occur in any or most of the above pathways. Each pathway presents varying levels of mutations in genes such as *BRAFV600E*, *NRAS*, *CDKN2A*, *TP53*, *PTEN*, *NF1* and *RET* [57]. In this classification system, less weight has been put on the clinical and histopathological criteria, emphasizing the importance of molecular criteria in melanoma classification and management.

One of the major studies describing genomic alterations in cutaneous melanomas (stages II to IV) was conducted by Akbani et al. (2015), leading to the establishment of a framework for genomic classification and the identification of immune-related prognostic biomarkers [40]. Using DNA, RNA and protein-based analysis, the authors classified cutaneous melanomas into four major subtypes based on the pattern of the most prevalent significantly mutated genes: mutant *BRAF*, mutant *RAS*, mutant *NF1* and Triple-WT (wild type), with each harboring relatively different UV signature, copy-number changes and structural arrangements [40]. In the same study, clustering analysis of 1500 most significantly changed transcripts identified three subclasses named as follows: “immune”, characterized by significant overexpression of immune-related genes; “keratin”, characterized by overexpression of genes associated with keratins and “MITF-low”, characterized by decreased expression of pigmentation and epithelial expression genes. The study found that there was a significant difference in the survival of patients with regionally metastatic tumors among the three clusters, with the “immune” subclass showing more favorable and the “keratin” subclass less favorable post-accession survival. This suggests the relevance of identified transcript expression subclasses for improved patient stratification.

Another study evaluated the accuracy of a gene expression profile (GEP) test in predicting metastatic risk in cutaneous melanoma patients [41]. Classifying the patients as either low or high risk, univariate and multivariate analyses of 31 GEP were found to be significant predictors of recurrence-free and distant metastasis-free survival. The GEP provided prognostic information to traditional staging and identified 70% of stage I and II patients who ultimately developed distant metastasis. The study suggests that the GEP test could be a useful tool for estimating an individual’s risk of recurrence and for considering adjuvant therapy in cutaneous melanoma patients.

A 2010 genomic study of stage IV melanomas identified four distinct subtypes of the tumor with different gene signatures related to immune response, pigmentation differentiation, proliferation or stromal composition genes [42]. The authors found a significant difference in mutations and deletions between the subtypes, with the proliferative subtype having a poor prognosis compared to the others. The clinical relevance of the subtypes was validated in an independent cohort of melanoma patients. This study highlights the importance of genome-based subtype classification for the effective and personalized management of melanomas.

Genomic analysis has also been able to identify key genetic mutations that can differentiate between melanomas from different body sites. For example, using a high-coverage whole-genome (WGS) sequencing analysis, distinct gene mutation processes and drivers including the landscape of non-coding mutations, paradoxical relationships between telomere maintenance gene mutations and telomere length have been found across cutaneous, acral and mucosal melanomas [58]. Cutaneous melanomas were found to be dominated by mutational signatures related to ultraviolet radiation exposure, while structural variants were responsible for most aberrations in acral and mucosal melanomas. These findings suggest that cutaneous melanomas are genetically different from acral and mucosal melanomas.

Considering the lack of consistency in melanoma classification using traditional approaches, and considering the above findings, the development of molecular classifiers that can be used alone or in combination with clinical and histological parameters has the potential to be a game-changing paradigm shift in the way we diagnose and manage melanoma.

#### 3.1.2. Cutaneous Squamous-Cell Carcinoma

Genomic studies have identified important mutational signatures and alterations in keratinocyte carcinomas, i.e., BCC and cSCC. Using whole-exome and whole-genome sequencing, important mutational signatures of primary and metastatic cSCC have been reported in the literature, providing unique resources to determine their biological significance for the maintenance and progression of the tumor [37,49]. Genome-wide association studies (GWAS) have also reported novel loci involved in cSCC development and keratinocyte differentiation, including those related to the JAK-STAT pathway [59,60,61].

One study used single-cell RNA sequencing with spatial transcriptomics and multiplexed ion beam imaging to define the cellular composition and architecture of cSCC and identified four tumor subpopulations, three recapitulating normal epidermal states, and a tumor-specific keratinocyte (TSK) population, a hub for intercellular communication [62]. Another study combined whole-exome analyses from 20 well-differentiated and 20 moderately/poorly differentiated tumors and found 16 genes including *TMEM51*, *GRHL2*, *ZZEF1* and *GMDS* with their mutation specific to the moderately/poorly differentiated subtype and six genes including *SULF1*, *ZNF528*, *NRCAM* and *FAT1* specific to the less aggressive well-differentiated subtype [37].

A study using targeted sequencing of 504 cancer-associated genes in lymph node metastatic cSCC samples reported a wide spectrum of oncogenic mutations affecting various genes and pathways. They identified specific mutations in oncogenic drivers (*TP53*, *CDKN2A*, *NOTCH1/2*) and pathways including RAS/RTK/PI3K, cell cycle and chromatin remodeling that were correlated with poorer patient outcomes [49]. Researchers have also identified and developed a 40-gene expression profile test that can predict the risk of metastasis risk in high-risk cSCC patients. The test which stratifies patients into three risk classes (low risk, high risk and highest risk of metastasis) can complement current staging systems for high-risk cSCC patients [50].

Large-scale DNA methylation study of premalignant actinic keratosis (AK) and cSCC has shown that the two lesion groups share the same methylation patterns resembling those of cSCC and exhibit features of stem-cell methylomes [63]. Interestingly, based on the keratin methylation patterns, the study also revealed the existence of two defined subclasses of AK and cSCC, one resembling healthy skin and the other one resembling tumor samples. A similar result was also found when the authors investigated TP63 gene methylation status in the lesions.

In SCCs, most genomic studies have focused on identifying gene biomarkers and classifiers against the currently accepted histopathological classification, leaving their unsupervised genomic classification, which has the potential to identify novel and clinically relevant subtypes, unexplored. Furthermore, the analysis of genomic classification and clustering has been limited to specific subtypes of tumors, without providing a comprehensive molecular classification and stratification system that covers the entire spectrum of lesions, including premalignant ones.

Nonetheless, taken together, the available literature described above suggests that implementation of molecular stratification and stratification techniques in cSCCs is poised to offer a more refined and accurate modality for discerning the various subtypes of cSCC, which in turn may lead to a more optimal approach to patient management, ultimately improving patient outcomes.

#### 3.1.3. Basal-Cell Carcinoma

Genomic and transcriptomic studies in BCCs have provided important information about its tumorigenesis. These include the identification of novel alleles and genetic mutations [60,64,65] and differentially expressed transcripts associated with tumor proliferation, migration, and apoptosis [66,67]. A 2015 gene expression study by Jee et al. identified three classes of BCCs (classical, SCC-like and normal-like BCCs) with distinct molecular characteristics, providing unique insights into the heterogeneous nature of the lesions [52]. For example, the authors found that the classical BCC subtype showed enriched activation of Wnt and Hedgehog signaling pathways, while the SCC-like BCC subtype was enriched with immune-response genes and oxidative stress-related genes. The classical BCC subtype was found to show prominent activation of metabolic processes, particularly fatty acid metabolism.

Exome and RNA sequencing analysis of infiltrative BCCs, a subtype of the lesion associated with poorer clinical outcomes, identified distinct molecular pathways and somatic mutations that lead to this tumor subtype. The study found that while infiltrative BCCs carry classical UV-induced mutational signatures, they display specific RNA expression profiles related to integrin and Wnt signaling [68]. The analysis of 13 BCC-related genes also found that superficial BCCs are significantly associated with *PTCH1* and *NOTCH1* mutations, with *NOTCH1* mutations being more frequent in lesions located on the trunk compared to the head/neck and extremities. The study provides further insights into the molecular alterations distinguishing between superficial and nodular BCCs [69]. Investigating superficial, nodular and morpheiform BCCs using cDNA microarrays, it has been reported that these BCC subtypes exhibit significant variation in gene expression patterns, particularly in genes associated with the MAPK pathway. Morpheiform BCCs were found to have unique upregulation of genes involved in response to DNA-damage stimulus, consistent with their more invasive phenotype [70]. Hierarchical clustering analysis of BCC samples based on RNA expression levels has also found a mixed cluster of high-risk and low-risk tumors with moderate upregulation of genes such as *SPHK1*, *MTHFD1* and *BMS1P20*. When clustering advanced versus non-advanced BCCs, a third group of lesions with no clear clustering with advanced and non-advanced tumors with moderate to highly moderate upregulation of genes including *COL1A1*, *COL1A2* and *COL3A1* were found [53].

Single-cell and spatial transcriptomics analysis have also been explored to highlight the molecular heterogeneity of BCC tissues in terms of tumor–stroma interactions at the invasive front of the tumor. Using single-cell RNA sequencing (scRNA-seq) transcriptomes and digital spatial profiling of infiltrative and nodular BCC samples, Yerly et al. (2022) identified 86 and 52 differentially expressed genes in the tumor and stroma, respectively [54]. The authors were also able to categorize the tumor nodular, tumor infiltrative, stroma nodular and stroma infiltrative areas of interests in BCCs based on their gene expression profiles. They also found that tumor cells at the invasive edge exhibit a collective migration phenotype, while nearby cancer-associated fibroblasts have extracellular matrix-remodeling features.

The above studies collectively suggest that BCC subtypes, as well as BCC regions in the same tumor from the same patient, exhibit molecular distinctions, and gaining insight into their genomic profiles and using them as molecular classifiers could aid in their improved management and clinical decision making.

### 3.2. Proteomic Classification in Skin Cancers

Proteomic analysis has been successfully employed to identify novel proteins and molecular differences that can discriminate between skin cancer subtypes, allowing clinicians to make informed curative or therapeutic decisions [46]. Unlike genes and transcripts, proteins are directly related to cell and tissue phenotypes [28,71]. Therefore, proteomic classification can provide a more comprehensive understanding of the molecular landscape of cancer, beyond what genomic analysis alone can offer. Proteins are the effectors of cellular function, and their activities are regulated by various post-translational modifications, such as phosphorylation, acetylation and glycosylation. These modifications can greatly affect protein function and contribute to cancer development and progression. Here, we summarize recent studies with a focus on the classification and stratification of melanoma, cSCC and BCC lesions at the proteome level. Also, a summary of the novel classes or groups of skin cancers with the potential to aid in their diagnosis is presented in Table 1a–c.

#### 3.2.1. Cutaneous Melanoma

In melanocytic lesions, mass spectrometry-based proteomics has shown the potential to perform better than histopathology in addressing diagnostic and classification questions, particularly the challenging lesions such as atypical Spitzoid melanoma [72] and congenital melanocytic nevus with proliferative nodules [73]. Also, comparing malignant melanoma with benign nevus, a proteomic-based model has shown an overall validated accuracy of 93% in classifying the lesions [74]. While the study’s findings may not be clinically significant due to the apparent differences in lesion size, depth of invasion and other morphologic changes between the two lesion groups, nonetheless, it shows the proteomics’ great potential for successful molecular classification of intermediate lesions as well.

One of the recent and most comprehensive proteomic studies of primary and metastatic melanoma was conducted by Betancourt et al., where more than 15,500 protein isoforms covering 65% of the total human proteome were quantified [75]. This study used histopathologically classified tumors to identify proteomic changes between the lesion groups, showing a very heterogeneous intertumoral and intratumoral disease. In this study, hierarchical clustering analysis using metastatic samples from four distinct body sites in the same patient identified different biological processes implicated in lesions from different sites. This study did not perform unsupervised classification analysis to investigate how proteomic data could classify the lesions based on their molecular landscape across the entire melanoma samples. The large number of samples in this study and the extensive proteome coverage provide a unique opportunity to investigate melanoma proteomic classification.

In another proteomic study using formalin-fixed paraffin-embedded (FFPE) tumor samples, the researchers identified six clusters of melanoma lesions using an unsupervised hierarchical protein clustering [43]. They found that proteins such as TRAF6 and ARMC10 were upregulated in clusters with shorter survival, while proteins like AIFI1 were upregulated in clusters with longer survival.

#### 3.2.2. Cutaneous Squamous-Cell Carcinoma

CSCC lesions, as well as their precursors, actinic keratosis (AK) and Bowen’s disease (BD) have been the subject of extensive proteomic investigation [16,17,28,51,76]. Studies have revealed protein biomarkers of AK, BD and cSCCs with different levels of tumor differentiation. It has been reported that differential expression of proteins exerting decreased apoptosis is associated with AK and cSCC lesions, while BD lesions are over represented by proteins associated with DNA damage repair pathways. On the other hand, proteins associated with alternatively spliced FGFR2 and Rho guanosine triphosphatase signaling are characteristics of cSCCs with different levels of tumor differentiation. Proteomic studies have also identified associations between cSCC lesion types and biological processes such as apoptosis and DNA damage repair.

While most of the studies use histopathology diagnosis as the basis for biomarker discovery analysis, interestingly, one study utilizing proteomic data as an independent classifier/clustering factor had identified primary and metastatic cSCC samples that were potentially miss-classified by histopathology assessment [76]. Another proteomic study comparing metastasizing and non-metastasizing primary cSCC lesions found that the expression of ANXA5 and DDOST proteins was associated with reduced time to metastasis [51]. Furthermore, a prediction model based on these two proteins showed better classification performance, with an accuracy of 91.2%, higher sensitivity and specificity compared to the existing clinical cSCC staging systems.

#### 3.2.3. Basal-Cell Carcinoma

In BCCs, histology-guided proteomic studies have suggested heterogeneous and chemically graded aggressive tumor islands [77], indicating the need for its molecular classification. Combining single-cell and spatial transcriptomics analysis has also been explored to highlight the molecular heterogeneity of BCC tissues in terms of tumor–stroma interactions at the invasive front of the tumor. Using single-cell RNA sequencing (scRNA-seq) transcriptomes and digital spatial profiling of infiltrative and nodular BCC samples, Yerly et al. (2022) identified 86 and 52 differentially expressed genes in the tumor and stroma, respectively [54]. Then, the authors were able to categorize the tumor nodular, tumor infiltrative, stroma nodular, and stroma infiltrative areas of interests in BCCs based on their gene expression profiles. They also found that tumor cells at the invasive edge exhibit a collective migration phenotype, while nearby cancer-associated fibroblasts have extracellular matrix-remodeling features.

Despite all the developments, however, molecular classification and stratification approaches have focused mainly on advanced lesions, leaving the early and intermediate precursors with overlapping clinical and histopathological features mainly unaccounted for. Nonetheless, the results from omics studies show the potential for a molecular classifier across all stages of skin cancers that can be used alone or in combination with the clinical and histopathology criteria to stratify patients more accurately and reliably.

## 4. Molecular Classifier for Therapeutic Stratification in Skin Cancers

Most patients with locally advanced or metastatic skin cancers who would not benefit from surgery or radiotherapy are treated with systemic therapies such as checkpoint inhibitor immunotherapy, chemotherapy and epidermal growth factor receptor (EGFR) inhibitors. While these therapies have changed the therapeutic landscape of skin malignancies, resistance to treatment remains challenging [14,15]. Currently, there is no reliable tool or guideline to suggest which patients would respond to this therapy and will therefore have longer disease-free survival. The development of resistance to therapy involves changes in the molecular landscape of tumor cells and the tumor microenvironment. Therefore, there is a critical need to develop molecular classifiers that can predict therapeutic responses, either alone or in combination with clinical and histological parameters. This will help match patients with the most effective therapy, and therefore, improve patient outcomes. In this context, we present a brief overview of recent molecular studies that have successfully identified subtypes of skin cancers based on their therapeutic responses. Also, a summary of recent molecular analyses in melanoma, cSCC and BCCs identifying novel classes or groups of lesions with the potential to predict clinical behaviors such as prognosis, survival, metastasis and response to therapy, is presented in Table 1a–c.

### 4.1. Cutaneous Melanoma

Similarly, primary and secondary resistance to immunotherapy in melanoma is also widespread, with only 22% of melanoma patients responding to ipilimumab and 40–45% to PD-1 immune checkpoint inhibitors [78]. In cutaneous melanomas, especially in the metastatic subtypes, proteomic classification has helped predict anti-PD-1 [46] or tumor infiltrating lymphocyte (TIL)-based immunotherapy outcomes in patients [46]. Proteomic analyses have also revealed site-specific response mechanisms to MAPK inhibitors in metastatic melanomas, suggesting site-specific treatments for increased efficacy and improved patient outcome [79]. Likewise, genomic and transcriptomic profiling has identified factors such as high *BRCA2 gene* mutational loads associated with increased response and upregulation of genes associated with mesenchymal transition, extracellular matrix remodeling and angiogenesis with increased resistance to anti-PD-1 therapy in metastatic melanomas [47]. Another study also demonstrated metastatic melanomas’ genomic heterogeneity in response to immune checkpoint blockade with higher CD8, CD45RO and PD-1, classifying the lesions into three responding groups [80].

In addition, transcriptomic analysis of stage I melanoma has been used to identify six classes of the lesion that can predict outcomes in patients, particularly those undergoing immunotherapy [48]. The reported transcriptome signatures in this study had shown to provide prognostic values comparable to that provided by sentinel node biopsy.

In a proteomic study comparing between primary and metastasis samples and clinical stages, and within immunotherapy and targeted therapy subgroups, the upregulation of the VEGFA-VEGFR2 pathway, RNA splicing, increased activity of immune cells, extracellular matrix and metabolic pathways were found to be positively associated with patient outcome [43]. The study highlighted the potential of proteomic profiling as an independent indicator of patients’ response to therapy and survival.

Despite all these findings, further research is needed to develop and clinically validate a more comprehensive molecular tool that could be used to predict all melanoma patients’ responses to therapy.

### 4.2. Cutaneous Squamous-Cell Carcinoma

Response to therapy in advanced cSCC patients remains poor. For example, 50% of advanced cSCC patients will not respond to the systemic therapy, those who respond will develop resistance over time [26] and 63% of the response to immunotherapy lasts only 16 weeks [81]. Patients with metastatic cSCC treated with EGFR inhibitor cetuximab obtains a complete response of 67% over 25 months, while those treated with cisplatin obtain a complete response of 17–22% over 14.6 months [81,82]. In addition, up to 12% of cSCC patients receiving systemic therapies develop Grade 3 or higher toxicities (severe and undesirable adverse events requiring hospitalization) [14,15]. Radiation therapy is also used as an alternative primary treatment for patients with metastatic cSCC who are not eligible for surgery. Veness and colleagues [83] reported that patients treated at Westmead Hospital (Sydney, Australia) between 1980 and 2000 for metastatic cSCCs, showed a 5-year local-regional recurrence and disease-free survival rates of 20% and 73% for surgery and radiotherapy, respectively. However, research in cSCC for developing a reliable molecular classifier to suggest which patients would respond to therapy and will therefore have longer disease-free survival is still lacking. Considering the high rate of resistance to therapy in advanced cSCCs, the development of molecular classifiers to stratify responding and non-responding tumors is highly desirable.

### 4.3. Basal-Cell Carcinoma

Good outcomes are observed when BCCs are diagnosed and treated early using current approaches such as Mohs micrographic surgery, radiotherapy and chemotherapy [84]. However, some BCCs do not respond to therapies, develop recurrence or progress to locally advanced and metastatic stages, and become difficult to treat [84].

Researchers have attempted to develop models for predicting response to treatment in BCCs to improve patient outcomes. For example, the initial response rate to Vismodegib, a small-molecule inhibitor, is about 50% in advanced and metastatic BCC [84]. A study on BCC patients who had received vismodegib found that a 20% reduction in tumor size after 3 months of treatment can predict an 82.76% chance of complete response, while a 67.7% reduction after 6 months can predict a 95.42% chance of complete response in patients [85]. This indicates that response to therapy is likely subject to the molecular composition of an individual patient’s tumor which can be exploited to develop useful tools to predict patients’ response to therapy.

Also, studies have identified biomarkers of resistance to Methyl-aminolevulinate photodynamic therapy (MAL-PDT), an effective treatment for the BCC [86]. It was found that patients with positive p53 immunostaining are 68.54 times more likely to respond to MAL-PDT therapy. A genomic sequencing study looking into genetic alterations leading to resistance to Hedgehog pathway inhibitors in BCC found that the majority of the resistant cases had Smoothened (SMO) gene mutations [55]. The study concluded that testing BCC samples for genetic mutations can be used as biomarkers to identify patients who are likely to develop drug resistance.

The above findings suggest that the development of a clinically validated comprehensive molecular classifier can be a powerful tool in predicting BCC patients’ response to therapy. Such a classifier can help clinicians select the most appropriate treatment for a patient based on the genetic and molecular characteristics of their disease.

Altogether, the above findings on melanoma, cSCC and BCC lesions suggest that the development of clinically validated molecular classifiers can be a powerful tool in predicting patients’ responses to a different therapy. Such a classifier can help clinicians to select the most appropriate treatment for a patient based on the genetic and molecular characteristics of their disease, facilitating precision medicine in skin cancer management.

## 5. Multi-Omics Classifier in Skin Cancers

It is well documented that the emergence of neoplastic properties in normal skin cells, tissue, and microenvironment results from changes in one or multiple layers of information from genes to transcripts, proteins and eventually phenotypes. On many occasions, using single data sources such as genes, proteins or imaging and clinical-histological patterns to build predictive and classification models is unreliable. Some clinical features of skin cancers are related strongly to histological features (tumor stages, for example), whereas others are associated with genomic, transcriptomic or proteomic alterations. Therefore, with advances in machine learning (ML) approaches and multi-modality data integration algorithms, “multi-omics” classification and prediction models have increasingly become popular in the past few years.

Deep learning models for multi-class classification have been found to outperform classification accuracy in cancers when only a single omic data is used [87]. Integrative analysis of omics data has identified previously unexplored subtypes of complex and heterogenous lesions such as ovarian and breast cancers, that are associated more significantly with the clinical outcomes than the established TCGA classification [88,89].

Considering skin cancers, the integration of WGS with transcriptome and methylome profiling has been able to classify melanoma patients and predict with up to 80% sensitivity in the validation set if one would respond to anti-PD-1 with or without anti-CTLA-4 therapy [44]. Furthermore, project HOPE (High-tech Omics-based Patient Evaluation) has used integrative whole-exome sequencing (WES) and gene expression profiling analysis to stratify melanoma patients who are considered to be good responders to anti-PD-1 therapy [45]. In a recent study involving single-cell RNA sequencing combined with spatial transcriptomics and multiplexed ion beam imaging, four distinct tumor subgroups of primary cSCC (basal, cycling, differentiating keratinocyte populations and tumor-specific keratinocyte population) were identified, uncovering the lesions’ spatial heterogeneity [62].

Although not in cutaneous melanoma, integrative clustering (iCluster) analysis of genomic, epigenomic and transcriptomic data in uveal melanoma has successfully identified four molecular subtypes of the lesion, each with a distinct multi-omics landscape, indicating its usefulness for accurate patient stratification and improved patient outcome [90]. In skin cancers, particularly melanoma, ML analysis of the imaging data (dermoscopy and histology images) has provided promising classification accuracy when comparing different lesion subtypes [91,92,93]. However, the combined ML analysis of imaging data together with other omic datasets has not been attempted in skin cancers. Combining artificial intelligence (AI)-generated data from hematoxylin and eosin (H&E) images of glioma and adenocarcinoma tumors with the genomic and/or proteomic data is reported to outperform the clinical prognosis approach [94,95]. This suggests that integrated analysis of imaging, genomic and proteomic data can also provide a useful source of information for the development of standardized multi-omics models for a reproducible and precise diagnostic and classification system in skin cancers.

Taken together, the above finding suggests that, in skin cancers, where accurate and effective diagnosis and therapeutic decisions are critical, a multi-omic classifier offers the potential to address the issues with current clinical and histopathological diagnostic and classification approaches.

## 6. Challenges and Opportunities with Molecular Classifiers

Given the challenges associated with the current clinical and histological classification and stratification of skin cancers, it is imperative to develop a reliable classification system that considers the molecular changes occurring in the lesions. There is growing evidence from cancers such as lung cancer [96], colorectal cancer [97,98], breast cancer [99] and leukemia [100] that molecular classifiers can significantly improve the identification of high-risk patients and can predict survival and therapeutic response compared with the conventional staging and stratification. Supported by the literature reviewed in this paper, molecular classifiers in skin cancers also have the potential to identify new classes of tumors that behave differently at the clinical levels based on the use of a robust set of biomarkers. This will provide clinicians with a standardized standalone or complementary tool that can predict the likelihood of recurrence, metastasis and response to therapy, allowing for personalized treatment approaches beyond established guidelines. Such pragmatic molecular classification tool has the potential to be used routinely in guiding surveillance and treatment tailored to specific subtypes of skin cancers.

Despite the available evidence, the development and implementation of robust molecular classifiers in skin cancer, significant challenges still exist. Advancements and successes in single-omics and multi-omics classifiers in cancers, including skin cancers, its clinical translation and implementation remain in infancy. The development of reliable molecular classifiers, especially multi-omics classifiers, requires large-scale and consistent datasets for data pooling to develop and validate the ML models [101]. There is no question that skin cancers are highly heterogenous, requiring comprehensive representation of all possible clinical and histological subtypes, different body sites, different geographical locations, times of biopsies, medications used, different ethnicity, history of treatment, tumor size and many more [101]. Overcoming these challenges will require a large-scale, multidisciplinary approach that includes independent validation and assessment in clinical trials to ensure the widespread use and effectiveness of the classifiers.

Additionally, the majority of molecular classifier studies have focused on a subset of advanced skin cancers with benign and early lesions where the outcome for early detection and management has largely been missed. For an inclusive and clinically useful molecular classification analysis, it is of utmost importance that samples encompassing premalignant lesions as well as early, intermediate and advanced lesions from across the carcinogenic process are included in the omics analysis.

For the diagnosis, classification and stratification of skin cancers, histopathology assessment of the excised lesion is often used as “ground truth” for final decision makings. For this reason, most omics studies have used supervised analysis approaches against histopathology as the gold standard to assess and define molecular classes. However, due to the lack of defined histological features in some cutaneous malignancies, the establishment of a final classifiers to be used as a reference to train ML classifiers can lead to the development of a flawed classifier system [102]. Therefore, the development of molecular classifiers approaches independent or partially independent of histopathology diagnosis may provide better and more clinically relevant information about the tumors’ behavior, response to therapy and overall outcome.

In addition, while there has been notable progress in the development and implementation of ML approaches for classification problems, there is more to be done to overcome the challenges that may limit the potential of unsupervised and multi-omics classifications for clinical implementation in complex and heterogeneous tumors. One of the main issues associated with the current mathematics and ML approaches is their performance and scalability attributed to the intrinsically large datasets in terms of features and samples that will require methods with the ability to decipher the intricate signals connecting different data modalities to clinically relevant problems [103,104]. Also, reviewed in a paper by Cai et al., data integration in multi-omics classification approaches produces only moderate to low consistency that needs to be mitigated [104]. The authors have also highlighted the issues with missing genuine biological relationships by using surrogate variables to describe variations between samples and recommend a gene-centric dynamic modelling approach in future multi-omics data integration.

## 7. Conclusions

In conclusion, recent advancements in omics approaches and the available evidence outlined in this paper indicate that molecular classifiers will allow for better classification and stratification of skin cancers beyond classical clinical and histological parameters, and that the multi-omics classifiers offer advantages by observing changes in the tumors at multiple layers of information.

## Figures and Tables

**Figure 1 cancers-15-04463-f001:**
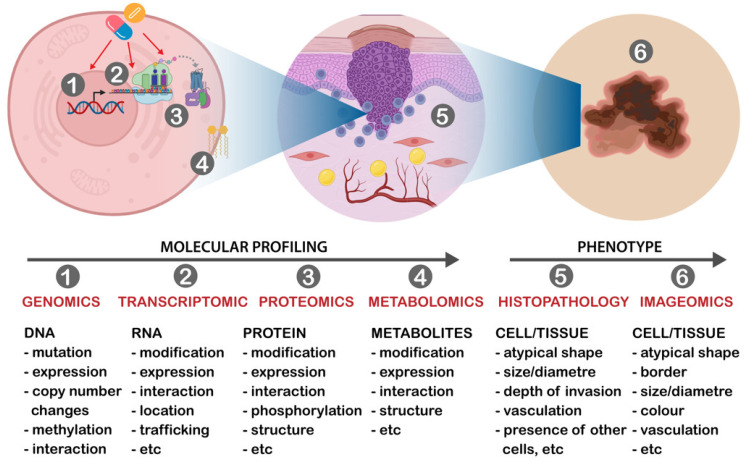
An overview of different layers of information that can be used to classify skin cancers. Like other tumors, the initiation and progression of skin cancers involve inherited or acquired mutations and alterations in genes (genomic) that are subsequently transcribed into RNAs with various modifications and alterations (transcriptomics). Upon the translation of RNAs, further changes are introduced in the cell at the protein (proteomics) and metabolite (metabolomics) levels which eventually permit the emergence of neoplastic properties in normal skin cells, leading to progression to premalignant lesions and advanced and aggressive carcinomas. Therapeutic agents will also infer their effects on tumors at these levels by interfering with cell functions. These molecular alterations are often, but not always, reflected in the tumor cell or tissue phenotype through changes in their size, shape, color, depth of invasion and microenvironment.

**Figure 2 cancers-15-04463-f002:**
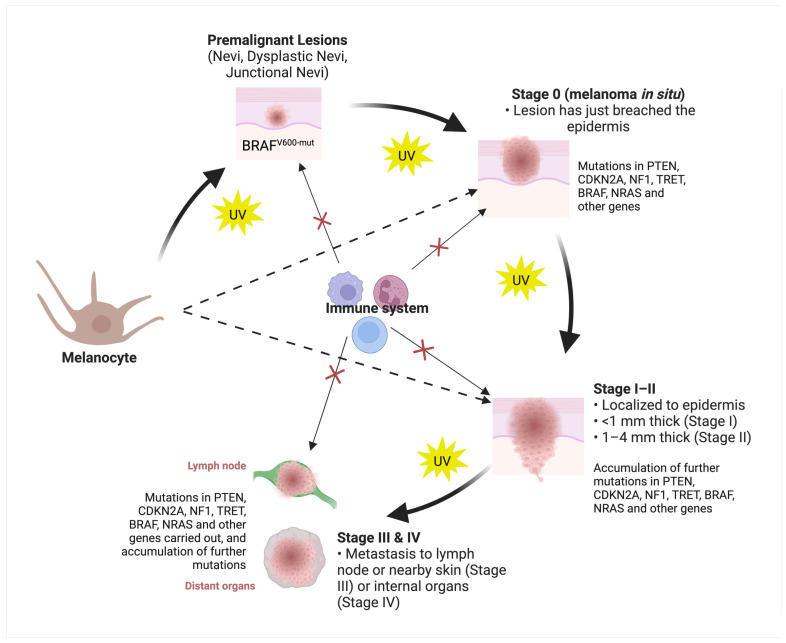
Overview of melanoma development and progression from melanocyte. This figure illustrates the sequential stages of melanoma development and progression. Melanocytes, sometimes influenced by hereditary gene mutations, tumor microenvironment such as immune inactivation and environmental elements, particularly UV exposure, can undergo genetic mutations, such as in the BRAF gene. This initiates the formation of premalignant melanoma lesions like nevi. As new and additional genetic mutations accumulate, along with post-translational modifications and other factors, these lesions may evolve, eventually encompassing the full epidermis layer (Stage 0 disease). Further genetic changes and alterations in the microenvironment can propel melanoma progression, leading to infiltration into the dermis layer (Stages I and II), with the potential for metastasis to nearby lymph nodes or distant organs (Stages III and IV). While the classical progression and pathogenesis involves distinct stages, some melanomas may deviate and develop de novo without progressing through the premalignant lesion stage (indicated by dashed arrow). This depiction aligns with the TNM staging system; × indicates inactivation or disruption of the immune system.

**Table 1 cancers-15-04463-t001:** Summary of genomic, transcriptomic or proteomic analysis in cutaneous melanoma, cutaneous squamous-cell carcinoma and basal-cell carcinoma identifying new classes or groups of lesions with the potential to assist in diagnosing or predicting clinical behaviors such as prognosis, survival, metastasis and response to therapy.

** *a. Melanoma* **				
**No**	**Features Attributing to the Classes/or “Omics” Analysis Used**	**Classes**	**Main Findings and Significance**	**Ref**
1	Genetic mutation information (major contributor), cumulative solar damage (CSD) and histology	1. Pathway I: superficial spreading melanoma (CSD, *BRAF p.V600* mutations) 2. Pathway II: lentigo malignant melanoma (*CSD; NF1, NRAS*, non-*p.V600E BRAF* mutations) 3. Pathway III: desmoplastic melanoma (CSD; inactivating *NF1*, promoting *NFKBIE*, and activating *MAPK* pathway mutations) 4. Pathway IV: spitz melanoma (no CSD; *HRAS* mutation, kinase fusions in *ROS1, NTRK1, NTRK3, ALK, BRAF, MET*, and *RET; CDKN2A* deletion, promoting TERT mutations) 5. Pathway V: acral melanoma (no CSD, CCND1, KIT, and TERT amplifications; BRAF, NRAS, and KIT mutations) 6. Pathway VI: mucosal melanoma (no CSD, copy number variations; KIT and NRAS mutations) 7. Pathway VII: melanoma arising in congenital nevi (no CSD (NRAS mutation in large congenital nevi; BRAF mutation in small to medium congenital nevi) 8. Pathway VIII: melanoma arising in blue nevi (no CSD (GNAQ, CYSLTR2, GNA11 and PLCB4 mutations; copy number aberrations in SF3B1 and EIF1AX) 9. Pathway IX: uveal melanoma (no CSD, GNAQ, GNA11, PLCB4, CYSLTR2, BAP1, SF3B1, and EIF1AX mutations)	Within this classification framework, reduced significance has been attributed to the clinical and histopathological factors, underscoring the elevated prominence of molecular criteria within the realm of melanoma classification and subsequent management of the tumor. This paradigm shift highlights the greater emphasis on discerning and utilizing molecular markers to inform the classification and comprehensive management of melanoma cases.	[38,39]
2	Whole-exome sequencing, DNA copy-number profiling, DNA methylation profiling and protein array expression profiling analysis	1. Mutant BRAF 2. Mutant RAS 3. Mutant NF1 4. Triple-WT (wild-type)	This study introduces a structured framework for genomic classification, identifying four distinct subtypes determined by the prevailing pattern of mutated genes.	[40]
3	Transcriptomic analysis	1. Immune: overexpression of immune-related genes) 2. Keratin: overexpression of genes associated with keratins 3. MITF-low: decreased expression of pigmentation and epithelial expression genes.	Regionally metastatic tumors in the “immune” subclass show more favorable and the “keratin” subclass less favorable post-accession survival, suggesting that transcript expression analysis will improve patient stratification.	[40]
4	Genomic analysis	1. Low risk of recurrence-free and distant metastasis-free survival 2. High risk of recurrence-free and distant metastasis-free survival	The risk of metastasis can be accurately predicted in 70% of stage I and II melanomas using 30-gene expression analysis, offering a useful tool for estimating individual’s risk of recurrence and for considering adjuvant therapy.	[41]
5	Genomic analysis	1. Immune response subtype 2. Pigmentation differentiation subtype 3. Proliferative subtype 4. Stromal composition subtype	There had been significant differences in mutations between the subtypes stage III and IV melanomas studied, with the proliferative subtype having a poor prognosis. Low expression of defined gene set associated with immune response was also found to be associated with poor outcome, highlighting the importance of genome-based subtype classification for personalized management of melanoma.	[42]
6	Proteomic analysis	Six clusters of melanomas based on their distinct proteomic profile showing different survival.	The study identified that proteins like TRAF6 and ARMC10 are linked to shorter survival, while AIFI1 is linked to longer survival. In the immunotherapy and targeted therapy groups, certain pathways and processes were linked to better patient outcomes, potentially aiding precision medicine.	[43]
7	Whole-genome, transcriptome, methylome and immune cell infiltrate analysis	1. Class 1: Respondents to anti-PD-1 therapy, with or without anti-CTLA-4 2. Class 2: Non-respondents to anti-PD-1 therapy, with or without anti-CTLA-4	Analysis of patients with advanced cutaneous melanoma undergoing anti-PD-1 therapy, with or without anti-CTLA-4 showed that response to immunotherapy is associated with high tumor mutation burden, neoantigen load, expression of IFNγ-related genes, programmed death ligand expression, low PSMB8 methylation and presence of T cells in the tumor microenvironment. A combined model involving tumor mutation burden and IFNγ-related gene expression predicted the response at AUC 0.79.	[44]
8	Whole-exome sequencing and gene expression profiling analysis	1. Class 1: Good responders to anti-PD-1 therapy. 2. Class 2: Non-responders to anti-PD-1 therapy.	Using integrative whole-exome sequencing and gene expression profiling analysis, melanoma patients with PD-L1 upregulation were found to be good responders to anti-PD-1 therapy.	[45]
9	Proteomic analysis	Classes of melanoma with different levels of aggressiveness	The expression of proteins such as nestin and vimentin could predict melanoma aggressiveness in different melanoma subgroups, allowing risk molecular stratification.	[46]
10	Genomic and transcriptomic analysis	1. Class 1: increased response to anti-PD-1 therapies 2. Class 2: increased resistance to anti-PD-1 therapies	Cactors such as high *BRCA2 gene* mutational loads are associated with increased response and upregulation of genes associated with mesenchymal transition, extracellular matrix remodeling and angiogenesis with increased resistance to anti-PD-1 therapy in metastatic melanomas.	[47]
11	Transcriptomic analysis	A total of 687 primary melanoma were categorized as classes 1 to 6, where classes 1 and 5 were typically thin and nonulcerated, classes 2 and 4 exhibited thicker characteristics. Class 3 and 6 tumors were the thickest and most frequently ulcerated. These six classes were significantly linked to mutation status: BRAF mutations were common in classes 1, 5, and 6, while NRAS mutations were frequent in classes 2, 3, and 4.	The performance of transcriptomic signatures in stage I melanoma showed similar indicator of prognosis when compared with sentinel node biopsy.	[48]
** *b: Cutaneous squamous-cell carcinoma* **				
**No**	**Features Attributing to the Classes/** **or “Omics” Analysis Used**	**Classes or Molecular Sub-Groups**	**Main Findings and Significance**	**Ref**
12	Whole-exome sequencing analysis	The study identified signatures of well-differentiated (six genes including SULF1, ZNF528, NRCAM and FAT1) and moderately/poorly differentiated (16 genes including TMEM51, GRHL2, ZZEF1 and GMDS) tumors.	This research elucidates the intricate molecular makeup of cSCC, uncovering driver genes, pathways, and mechanisms linked to the formation of well-differentiated and moderately/poorly differentiated tumors.	[37]
13	Targeted genomic analysis	This study identifies metastatic cSCC patients with overall good or poor survival.	Substantiates the connection between mutations in chromatin-modifying genes or mutations involving chromatin modifiers in combination with RAS/RTK/PI3K and unfavorable outcomes.	[49]
14	Genomic analysis	1. Class 1: patients with low risk of metastasis 2. Class 2: patients with high risk of metastasis 3. Class 3: patients with highest risk of metastasis	Using a 40-gene expression test, the risk of metastasis can be predicted in primary cSCC patients, complementing current staging systems for high-risk patients.	[50]
15	Proteomic analysis	Class 1: patients with high risk of metastasis Class 2: patients with low risk of metastasis	Primary cSCC lesions with higher levels of ANXA5 and DDOST proteins is associated with reduced time to metastasis. A prediction model based on these proteins showed a classification performance with an accuracy of 91.2% and higher sensitivity and specificity compared to the existing clinical cSCC staging systems.	[51]
** *c: Basal-cell Carcinoma* **				
**No**	**Features Attributing to the Classes** **or “Omics” Analysis Used**	**Classes**	**Main Findings and Significance**	**Ref**
16	Transcriptomic analysis	1. Class 1: classical BCC 2. Class 2: SCC-like BCC, 3. Class 3: normal-like BCC	Every subgroup exhibited specific molecular traits, offering distinct understanding into the diverse features of these lesions. For instance, the classical BCC subtype demonstrated heightened engagement of Wnt and Hedgehog signaling pathways, whereas the SCC-like BCC subtype displayed enrichment in genes tied to immune responses and oxidative stress. Additionally, the classical BCC subtype exhibited marked activation of metabolic pathways, with a notable emphasis on fatty acid metabolism.	[52]
17		Hierarchical clustering analysis of BCC samples based on RNA expression levels has also found a mixed cluster of high-risk and low-risk tumors with moderate upregulation of genes such as *SPHK1*, *MTHFD1* and *BMS1P20 *. When clustering advanced versus non-advanced BCCs, a third group of lesions with no clear clustering with advanced and non-advanced tumors with moderate to highly moderate upregulation of genes including *COL1A1*, *COL1A2* and COL3A1 were found.		[53]
18	Single-cell and spatial transcriptomics analysis	The authors identify the tumor nodular, tumor infiltrative, stroma nodular and stroma infiltrative areas of interests in BCC, each with a distinct genomic profile.	This study reveals distinct gene expression differences between tumor and stroma cells in infiltrative and nodular BCC samples, and notes that invasive edge tumor cells exhibit collective migration phenotype, while nearby fibroblasts remodel the extracellular matrix.	[54]
19	Transcriptomic and whole exome analysis	Class 1: BCCs resistant to vismodegib treatment Class 2: BCCs sensitive to vismodegib treatment	The study discovered SMO mutations in half of the resistant BCCs, demonstrating their role in sustaining Hedgehog signaling despite SMO inhibitor (vismodegib) treatment. These findings highlight SMO gene mutations as significant contributors to resistance. Consequently, the research suggests that screening for genetic mutations in BCC samples could serve as a useful method for predicting drug resistance in patients.	[55]

## Data Availability

Not applicable.

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
