# Peer review of "Molecular Classifiers in Skin Cancers: Challenges and Promises"

_cancers, 2023, doi:10.3390/cancers15184463_

Round 1

Reviewer 1 Report (Previous Reviewer 1)

Dear authors,

The manuscript was significantly improved after adjusting the manuscript according to the reviewers' suggestions.

I found Figure 2 confusing. In the text (paragraph starting at line 186), the authors report that one of the major studies was conducted by Akbani et al. However, figure 2 does not mention this study. Also, the explanation of the studies described in Figure 2 should be better developed. On line 194, for example, three clusters are described, but it is not explained what cluster “keratin” is.

Figure 2 legend: “… Details information about each study is provided in provided in the manuscript text.”

In some parts of the text, the authors describe only cutaneous carcinomas, but the manuscript also discusses melanoma (e.g. line 411 – “Molecular Classifier for Therapeutic Stratification in Cutaneous Carcinomas).

Author Response

Response to Reviewer 1 Comments

Point 1:

I found Figure 2 confusing. In the text (paragraph starting at line 186), the authors report that one of the major studies was conducted by Akbani et al. However, figure 2 does not mention this study. Also, the explanation of the studies described in Figure 2 should be better developed. On line 194, for example, three clusters are described, but it is not explained what cluster “keratin” is.

Response 1: We thank the reviewer for the comments provided.

We realised that the citation in Figure 2 was incorrect. The two classifiers in Figure 2 were indeed related to the studies by Akbani et al., which are now fixed.

Also, in the main text, we have provided detailed information about the molecular subclasses of melanoma described in Figure 2.

Expanded text:

In the same study, clustering analysis of 1,500 most significantly changed transcripts identified three sub-classes named "immune" characterized by significant overexpression of immune-related genes, "keratin", characterized by overexpression of genes associated with keratins, and "MITF-low", characterized by decreased expression of pigmentation and epithelial expression genes. The study found that there was a significant difference in the survival of patients with regionally metastatic tumours among the three clusters, with the “immune” subclass showing more favourable and the “keratin” subclass less favourable post-accession survival. This suggests the relevance of identified transcript expression subclasses for improved patient stratification.

 Point 2:

Figure 2 legend: “… Details information about each study is provided in provided in the manuscript text.”

Response 2: We thank the reviewer for bringing this paper to our attention. The sentence has been fixed.

Detailed information about each study is provided in the manuscript text.

Point 3:

In some parts of the text, the authors describe only cutaneous carcinomas, but the manuscript also discusses melanoma (e.g., line 411 – “Molecular Classifier for Therapeutic Stratification in Cutaneous Carcinomas).

Response 3: We sincerely appreciate your valuable feedback regarding our manuscript. We acknowledge the controversy and confusion surrounding the definition of carcinoma "cancers arising from epithelial tissue”. In order to address this concern and to avoid any potential confusion, we have replaced the term "cutaneous carcinoma" with "skin cancer" throughout the manuscript. This broader term encompasses squamous cell carcinomas, basal cell carcinomas, and melanoma. By adopting this terminology, we aim to encompass the full spectrum of skin cancers without inadvertently misclassifying melanoma as a subtype of carcinoma.

Reviewer 2 Report (Previous Reviewer 2)

The authors addressed my comments and also greatly improved the flow and content of the manuscript.

Author Response

We sincerely appreciate the time and effort devoted by the reviewer to reviewing this manuscript and providing valuable feedback that has contributed to its improvement. Your insightful comments have undoubtedly enhanced the quality of our work, and we are thankful for your valuable contributions.

Reviewer 3 Report (Previous Reviewer 3)

I had previously reviewed this manuscript, and in this new version, the authors made a great effort to resolve all suggestions and questions. The redaction was improved, and now that I consider this manuscript acceptable, I recommend it without changes.

Dear editor.

Thank you again for allowing me to review this manuscript that I previously reviewed. The authors made a great effort to improve the manuscript, and they responded to all my suggestions, which I recommend without changes

Author Response

We sincerely appreciate the time and effort devoted by the reviewer to reviewing this manuscript and providing valuable feedback that has contributed to its improvement. Your insightful comments have undoubtedly enhanced the quality of our work, and we are thankful for your valuable contributions.

Reviewer 4 Report (New Reviewer)

In this review the authors  introduce all the latest advances in the omics classification of cutaneous carcinomas. The article is well written and it is improved beacuse the authors addressed all the reviewers comments.

Author Response

We sincerely appreciate the time and effort devoted by the reviewer to reviewing this manuscript and providing valuable feedback that has contributed to its improvement. Your insightful comments have undoubtedly enhanced the quality of our work, and we are thankful for your valuable contributions.

This manuscript is a resubmission of an earlier submission. The following is a list of the peer review reports and author responses from that submission.

Round 1

Reviewer 1 Report

The manuscript summarizes the omics changes and their importance in cutaneous carcinomas.

This is an important field to be better explored to improve the clinical management of these patients.

Minor corrections should be made:

Line 37: reference 1 is a study about lung adenocarcinoma. It does not mention cutaneous carcinomas

Figure 1: copy number chnages

Reviewer 2 Report

Azimi and Fernandez-Penas summarized the current landscape of cutaneous carcinomas highlighting the efforts to characterize these cancers by omic technologies. The paper is well written and easy to follow.

Minor revisions:

For genomic profiling. I think the authors could mention the paper from Hayward et al Nature 2017 as a genomic classifier for different subtypes of melanoma.

For reference 68, the final model was using TMB and the IFG gamma signature to achieve 89% in the training set and 80% sensitivity in the validation set.

Another important paper for multi-omic melanoma classification would be from Newell F 2022 Cancer cell 

Perhaps the authors could also mention another study for BCC in spatial omics  Yerly L et al 2022 Nature communications. 

Reviewer 3 Report

The manuscript made by Azimi A et al., it is related to a review of several molecular pathways in different types of carcinomas, including melanoma. Despite the manuscript is interesting, it is confused and disorganized. Authors try to explain the function of genomics related to prognosis and diagnosis. However, the authors describe it in a confusing way, intermingling concepts between cutaneous carcinoma, melanoma, and other types of carcinomas. Also, some of citations are not enterally related with the manuscript that authors cited, for example, line 146 to 147, "NRAS mutations...." they describe several molecular related with the diagnosis comparing it with the histopathology diagnosis, when more than one of these molecules are not related with diagnosis in melanomas.
Authors describe Figure 1, but this figure does not have any relationship with the text of the manuscript, authors need to relate the figure with the text in overall this manuscript despite is interesting this is intermingled confused, and subtitles are not related enterally with the text of the manuscript.
This manuscript needs a deeper review and better organization. Authors need to include tables and figures in the text.

Reviewer 4 Report

The authors provide a review of advances in multi-omics techniques in diagnosis of cutaneous carcinomas.

1. One of the key issues that I have with this review is that authors have attempted combining studies for basal cell (BCC) and cutaneous squamous cell carcinomas (cSCC) and melanomas. These are all skin cancers but in terms of diagnosis and developmental criteria these are quite different. There is no evidence of basal cell carcinomas having been misdiagnosed. In fact one of the key skin cancer that is cutaneous T-cell lymphoma (CTCL) and its various subtypes are often misdiagnosed as dermatitis or psoriasis which the authors failed to include in this comparison. 

2. Due the inclusion of various diverse skin carcinomas the authors are unable to provide key examples where the latest omics techniques improved the diagnosis except in melanoma but at the same time importance of genetic testing in melanoma is well documented. 

3. The authors also fail to provide a key arguments as to how further developments of omics is going to improve the diagnosis of cutaneous carcinomas. 

Its my recommendation that the authors reconsider the aim of this review and maybe focus on fewer cutaneous carcinomas or rearticulate the sections with focus on how certain genomic or transcriptomic sequencing development improved diagnosis and therapy options in each of their selected carcinomas.